# Effects of CCL20/CCR6 Modulators in a T Cell Adoptive Transfer Model of Colitis

**DOI:** 10.3390/ph18091327

**Published:** 2025-09-04

**Authors:** Marika Allodi, Lisa Flammini, Carmine Giorgio, Maria Grazia Martina, Francesca Barbieri, Vigilio Ballabeni, Elisabetta Barocelli, Marco Radi, Simona Bertoni

**Affiliations:** Department of Food and Drug, University of Parma, Parco Area delle Scienze 27/a, 43124 Parma, Italy; marika.allodi@unipr.it (M.A.); lisa.flammini@unipr.it (L.F.); carmine.giorgio@unipr.it (C.G.); mariagrazia.martina@unipr.it (M.G.M.); francesca.barbieri@unipr.it (F.B.); vigilio.ballabeni@unipr.it (V.B.); elisabetta.barocelli@unipr.it (E.B.)

**Keywords:** CCL20/CCR6, intestinal inflammation, adoptive transfer colitis, CCR6 antagonist

## Abstract

**Background/Objectives**: IBDs are chronic relapsing inflammatory intestinal disorders whose precise etiology is still only poorly defined: critical for their pathogenesis is the CCL20/CCR6 axis, whose modulation by small molecules may represent an innovative therapeutic approach. The aim of the present work is to test the potential efficacy of two molecules, **MR120**, a small selective CCR6 antagonist, active in TNBS- and chronic DSS-induced murine models of intestinal inflammation, and its derivative **MR452**, a well-tolerated agent endowed with improved anti-chemotactic in vitro properties, in the adoptive transfer colitis model. To the best of our knowledge, this is the first attempt to use adoptive transfer colitis to test modulators of the CCL20/CCR6 axis. **Methods and Results**: The induction of colitis in immunocompromised mice receiving CD4^+^CD25^−^ T cells i.p. resulted in a moderate inflammation and was met with limited protective responses following daily subcutaneous administration of **MR120** or **MR452** for 8 weeks. Both compounds significantly reduced colonic myeloperoxidase activity, and **MR452** also lowered CCL20 levels in the gut, but they failed to prevent the increase in the Disease Activity Index, colon wall thickening, and macroscopic inflammation score. **Conclusions**: Our findings suggest that, despite the beneficial effects played by **MR120** against subacute TNBS- and chronic DSS-induced colitis, the pharmacological targeting of the CCL20/CCR6 axis in the adoptive transfer model has a negligible effect in ameliorating the IBD-like phenotype driven by the altered intestinal immune homeostasis and by the disrupted function of immune-suppressive Treg cells.

## 1. Introduction

IBDs are a complex of chronic relapsing inflammatory disorders of the intestine whose incidence and prevalence are increasing steadily all over the world and whose impact on everyday life is dramatic [1]. The pharmacological therapy currently available consists mainly of traditional drugs (aminosalicylates, corticosteroids, immunosuppressive agents) and monoclonal antibodies. However, the tolerability and efficacy of current drugs are limited, and a substantial number of IBD patients fail to respond or to fully remit, while respondents can lose response over time [2]. Therefore, having drugs able to maintain high efficacy and limited side effects in all patients is an important unmet need.

Although the precise etiology of IBD is still only poorly defined, a central role in its pathogenesis is played by the dysregulation of the gut barrier function and by the continuous recruitment of leukocytes from the circulation to inflamed tissues [3]. This is proved by the therapeutic efficacy of strategies interfering with gut homing of effector leukocytes, as in the case of anti-human α4β7 vedolizumab [4]. Crucial for IBD development is also the disrupted balance between CD4^+^Th17 (pro-inflammatory) and Treg (regulatory) cell populations, witnessed by the elevated IL-17 blood levels and intestinal infiltration of Th17 cells in IBD patients [5,6]. To this process, the interaction of CCL20 (CC chemokine ligand 20) with its sole receptor CCR6 could make a relevant contribution. CCL20, expressed constitutively by lymphoid tissues and epithelial cells, is released by a variety of other immune and non-immune cells in inflammatory conditions [7] and promotes the migration of CCR6-expressing leukocytes, CD4^+^ (including Treg and Th17 cells) and CD8^+^ T cells, B cells, dendritic and antigen-presenting cells, and activated neutrophils [7]. The participation of the CCL20/CCR6 axis to IBD pathogenesis is supported by various observations: the altered expression of both CCR6 and its partner chemokine in the colonic mucosa and serum of IBD patients [8,9] and the identification of their coding genes as susceptibility genes for IBD [10]; the ability of the anti-CCL20 neutralizing antibody to protect against TNBS-induced colitis, preventing neutrophils and T cells infiltration in the colon [11]; the lower severity of DSS-induced colitis in CCR6-knockout mice [12]; and the interference played by CCL20 signaling in human Treg differentiation, instead promoting Th17 lineage [13]. Overall, these data highly suggest that the CCL20/CCR6 axis is directly involved in IBD pathogenesis and its modulation by small molecules may represent an innovative and useful therapeutic approach for IBD patients.

Moving from these grounds, our group has recently developed compound **MR120** (Figure 1), a small and selective CCR6 antagonist whose anti-inflammatory efficacy has been proved by us in three different models of intestinal inflammation: subacute TNBS-induced colitis, considered a preclinical model of Crohn’s disease [14], zymosan-induced peritonitis, reproducing acute inflammation associated with strong native immune responses [14], and chronic DSS-induced colitis, a model able to more closely mimic the typically alternating phases of remission and exacerbation of human IBD [15]. The efficacy of **MR120** as a CCR6 antagonist able to mitigate the inflammatory cell recruitment was also confirmed by Yoo et al. in a model of murine acute ischemic kidney injury [16]. In general, the ability of this small molecule to prevent the inflammatory responses and the local and systemic neutrophil infiltration clearly supported the modulation of the CCL20/CCR6 axis as an advantageous strategy to limit intestinal and gut-derived systemic inflammation. In a following study, centered on the optimization of **MR120**, we designed and synthesized a focused collection of derivatives, among which **MR452** emerged as a well-tolerated anti-chemotactic agent against the in vitro CCL20-induced migration of human peripheral blood mononuclear cells, and also performed better than the reference compound **1** developed by Takeda Pharmaceutical (Osaka, Japan) [17] (Figure 1) [18].

Building on these premises, the present study aims to investigate the potential protective action of **MR120** and **MR452** in the T cell transfer-induced chronic colitis model, concurrently evaluating for the first time its suitability as an in vivo preclinical model to test CCR6 antagonists. Adoptive transfer colitis, firstly conceived by Powrie [19] and based on the injection of CD45Rb^high^ subset of CD4^+^ T cells to syngeneic immunodeficient, SCID or Rag1^−/−^, mice, has become a predictive model of chronic colitis ensuing from the perturbation of intestinal immune homeostasis, and is considered similar to CD in terms of histopathology and gene expression patterns [20]. An alternative but less expensive colitis model for pharmacological testing of new drug candidates, in terms of required technical expertise and high yield of transferable cells for colitis induction, was developed by Kjellev et al. and was based on the transfer of CD4^+^CD25^−^ T cells, a CD4^+^ subtype lacking Treg cells, to syngeneic immunodeficient mice [21]. This experimental model of intestinal inflammation, possibly ensuing from the absence of CD4^+^ Treg cells and from the impairement of intestinal immune homeostasis [22], could therefore represent a supplementary tool to investigate the potential beneficial action of CCR6 antagonists **MR120** and **MR452** in intestinal inflammation, given the expression of CCR6 receptors on immune cells and their role in their recruitment towards the inflamed sites.

## 2. Results

### 2.1. Effects on Inflammatory Responses

Intraperitoneal infusion of CD4^+^CD25^−^ T lymphocytes slowly worsened the clinical conditions of vehicle-treated C.B.-17 SCID mice, as demonstrated by the increasing DAI score over the weeks (C), while no changes were detected in immunocompromised mice that were i.p. infused with pure naïve CD4^+^ T cells (S). Vehicle-treated colitic mice (C) showed a marked deterioration of the health status from the fifth week until the end of the experimental protocol (* *p* < 0.05 vs. S). The treatments with **MR120** and **MR452** were not able to improve the general health status of mice, showing DAI curves comparable to those of the (C) group (Figure 2).

The development of adoptive transfer colitis did not lead to colon shortening or relevant splenomegaly (Figure 3a,b), but significantly augmented colon thickness and caused a remarkable increase in the macroscopic score in all inflamed experimental groups (* *p* < 0.05 vs. S) (Figure 3c,d). The administration of **MR120** and **MR452** to colitic mice was not able to reduce the local and systemic inflammatory parameters, but both colon thickening and the severity of lesions were comparable with those of vehicle-treated mice (C), as reported in Figure 3c,d.

As regards colonic MPO activity, it resulted markedly higher in vehicle-treated colitic C mice compared to S mice (* *p* < 0.05 vs. S), and both treatments were able to reduce neutrophil oxidative activity (^#^
*p* < 0.05 vs. C) (Figure 4).

### 2.2. Effects on T Lymphocytes

The flow cytometry analysis performed on mesenteric lymph nodes (MLNs) and the spleen revealed an increase in T lymphocytes in MLNs of colitic mice (C) (Figure 5a), different from their absence in the spleen (Figure 5b). The increase in T lymphocytes percentage during the inflammatory process was particularly marked in vehicle- and **MR120**-treated mice compared to sham mice (* *p* < 0.05 vs. S) (Figure 5). A small but not significant reduction in T lymphocytes in **MR452**-treated mice compared to vehicle-treated mice was seen. Within CD3^+^ lymphocytes, CD3^+^CD4^+^ cells represented the predominant subpopulation, showing comparable percentages in all the experimental groups (Table 1).

### 2.3. Effects on Inflammatory Cytokines

In vehicle-treated colitic mice, the production of inflammatory cytokines in the colon was slightly increased with respect to sham mice, especially in the case of CCL20 (* *p* < 0.05 vs. S), while IL-1β and IL-6 levels did not markedly increase, possibly due to the slow onset of inflammation (Figure 6). As regards the effects of the small molecules, only the treatment with **MR452** was able to significantly diminish the levels of CCL20 (^#^
*p* < 0.05 vs. C) (Figure 6a), while no changes were produced by either compound on IL-1β or IL-6 colonic release (Figure 6b,c).

As regards the determination of mRNA levels of inflammatory cytokines in the colon, the gene transcription for *CCL20* and *IL-17* was comparable in all the experimental groups (Figure 7a,b), although a small, non-significant increase in IL-*17* mRNA levels could be detected in C mice with respect to S ones (*p* = 0.054), a trend counteracted by **MR120** and **MR452** (Figure 7b). As regards *INF-γ*, gene transcription was moderately increased in colitic mice, but the small molecules did not show a marked decrease in mRNA levels of the cytokine with respect to C mice (Figure 7c). Only a small but non-significant reduction in INF-*γ* in **MR452**-treated mice compared to C mice was seen.

## 3. Discussion

Accumulating evidence indicates that a central role in IBD pathogenesis is played by the dysregulation of immune response against luminal antigens through a defective epithelial barrier and under the influence of various environmental triggers [23]. Well aware that no single model can capture the complexity of human IBD, several different murine experimental colitis protocols have been conceived, each of which was based on a specific pathogenetic mechanism, including models with epithelial barrier defects (i.e., DSS-induced colitis [24]), models with excessive effector T cells responses (i.e., TNBS-induced colitis [25]), or with regulatory and effector T cell imbalance (adoptive transfer colitis [19]). The poor predictivity of human diseases in mouse models is a well-known challenge, often referred to as the “mouse trap” or the “translational gap” [26], and corroborating findings across multiple models might provide key insight in choosing the more suitable approach to test IBD drug candidates with a new mechanism of action (i.e., inhibition of the CCL20/CCR6 axis).

Following the application of subacute TNBS-induced colitis and chronic DSS-induced model, the benzofuran derivative **MR120**, functioning through the modulation of the CCL20/CCR6 axis, has progressively unfolded as a compound capable of preventing the flogistic responses and the local and systemic neutrophil infiltration, underscoring gut inflammation. Additionally, the efficacy of this compound at the same dosage (1 mg/Kg) has been emphasized by Yoo et al. in a model of murine acute ischemic kidney injury [16], further supporting its potential in immune-related inflammation. The favorable results on efficacy and safety obtained by **MR120** boosted the design and synthesis of a focused collection of novel analogs based on the same chemotype. Among these, **MR452** emerged as the best candidate, demonstrating high tolerability in PBMCs and exhibiting enhanced in vitro anti-chemotactic activity [18].

In the present study, we evaluated **MR120** and **MR452** in a direct comparative analysis using T cell adoptive transfer colitis, which to our knowledge has not previously been applied to CCL20/CCR6 modulators, with the aim of complementing data from previous chemical models and obtaining a more comprehensive view of how different in vivo models respond to this drug class [23]. The T cell transfer model we applied was based on the transfer to syngeneic immunocompromised SCID mice of CD4^+^ T cells lacking CD4^+^CD25^+^ T cell subtype, an immune-suppressive subpopulation first characterized by Read et al. as expressing the same phenotype of Treg cells [27].

In our conditions, the intraperitoneal infusion of CD4^+^CD25^−^ cells to immunocompromised mice evoked only a moderate colitis over the 8 weeks of the investigation: a gradual worsening of the clinical status, characterized by a slow increase in DAI score and splenomegaly, was detected in comparison with sham mice, which received CD4^+^ cells in toto. Consistent with the fact that the inflammatory process, induced by the T cell transfer in the colitis model, specifically affects the intestinal district, thickening of the wall, enhanced oxidative activity of neutrophils and increased mucosal damage in the colon were observed, supporting the results presented by Kjellev [21]. As regards lymphocyte responses, despite an almost complete absence of T lymphocytes in the spleen, a significant rise in the number of T lymphocytes, mainly represented by CD3^+^CD4^+^ cells, was registered in MLNs of colitic mice, suggesting the contribution of these cells, lacking Treg type, in triggering intestinal inflammation. Indeed, along with the mucosal lesions, a moderate increment of mature CCL20 levels and the augmented transcription of *INF-γ* gene were revealed in the colonic tissue, confirming the typical features of T cell adoptive transfer colitis [28,29].

In this in vivo model, the administration of **MR120**, at the same dose that was effective against subacute TNBS- and chronic DSS-induced colitis (1 mg/Kg), and of its analog **MR452** showed a weak local protective effect, mitigating the oxidative action of neutrophils in the colon, feebly reducing *IL-17* mRNA levels, and, only in the case of **MR452**, counteracting the increase in CCL20 production. Neither of the two small molecules exhibited systemic beneficial effects, nor was any disruption observed in the recruitment of CD3^+^CD4^+^ lymphocytes to the mesenteric lymph nodes. These results seem to indicate that the interference with the CCL20/CCR6 axis by **MR120** and **MR452** has negligible effects in hindering the gut homing of effector lymphocytes and in relieving the inflammatory role of CD4^+^CD25^−^ T cells.

Interesting reports from the literature showed that the transfer of CCR6^−/−^ naïve T cells into *Rag2*^−/−^ mice led to more aggressive inflammation with respect to the infusion of wild-type cells, suggesting that CCR6^−/−^ Treg cells have less suppressive action compared to WT Treg cells [28]. This observation apparently undermines the rational underlying our investigation with CCR6 antagonists in the same model. However, it is widely acknowledged that different phenotypes may ensue from targeting a protein function, either via small molecule inhibitors or gene knock out [30]: accordingly, the fact that CCR6 deficiency via gene knockout of donor T cells aggravates colitis in the T cell adoptive transfer model does not univocally mean that CCR6 antagonists would worsen colitis. We could not exclude that both genetic deletion and CCR6 antagonists may exert off-target effects, which could differ from each other. Moreover, genetic deletion might trigger compensatory changes in other genes during mice development, potentially influencing the role of CCR6 receptor in intestinal inflammation in the adult mice that not necessarily correspond to the in vivo effects evoked by CCR6 inhibitors. Indeed, the results collected in our experimental conditions suggest that the interference with the CCL20/CCR6 axis, achieved not through *CCR6* gene deletion but via pharmacological modulation, does not appear to exacerbate the colitis induced by the lack of Treg cells, while only colonic MPO reduction was observed, not comparable with the systemic and local beneficial effects previously shown by **MR120** on DSS- and TNBS-induced colitis models. Several reasons may account for these limited advantages: besides the fact that a single dose was tested, while higher doses may produce stronger, either positive or negative, effects, the slow and progressive nature of the adoptive transfer model, in which colitis typically develops over 4–6 weeks and is primarily driven by T cell-mediated immune dysregulation, emphasizes as critical the timing of intervention. While our use of a preventive treatment might seem optimal for demonstrating anti-IBD drug efficacy, as supported by Lindebo Holm et al. [31], other studies suggest that later interventions can be more advantageous [32]. Furthermore, the moderate epithelial injury and chronic inflammatory remodeling may underestimate the compounds pharmacological effects. In contrast, the TNBS- and DSS-induced colitis models—which feature rapid onset and early chemokine-driven immune cell recruitment—appear more responsive to CCR6-targeted intervention. These findings highlight the importance of disease evolution, immune context, and treatment timing in the preclinical evaluation of compounds targeting CCL20/CCR6 axis, and raise concerns about the sensitivity and appropriateness of the T cell transfer model for identifying drug candidates acting through this pathway.

In conclusion, our findings show that, despite the beneficial effects played by **MR120** against models directly compromising excessive effector T cells responses and epithelial barrier integrity (subacute TNBS- and chronic DSS-induced colitis [14,15]), the pharmacological targeting of the CCL20/CCR6 axis in the adoptive transfer model seems to have a negligible action in ameliorating the IBD-like phenotype. This could suggest that the ability to reduce the neutrophil oxidative action, exhibited by **MR120** in the three different colitis models and shared by **MR452** in the adoptive transfer, is apparently not sufficient to contrast the systemic effects and colonic epithelial injury in a model of IBD driven by the altered intestinal immune homeostasis and by the disrupted function of immune-suppressive Treg cells: here, innate responses are probably overridden by the involvement of adaptive cells. Future studies will be pivotal to have a deeper insight into the impact **MR120** and **MR452** may have on the unbalance between gut CD4^+^ Treg and Th17 lymphocytes and on their interaction with innate immune cells.

## 4. Materials and Methods

### 4.1. Animals

Male Balb/c (6–8 weeks old, weighing 25–30 g) and C.B-17 SCID (6 weeks old, weighing 18–25 g) mice (Charles River Laboratories, Calco, Italy), were housed five per cage and maintained under standard conditions at our animal facility (Balb/c mice) or bred under SPF conditions in a ventilated cabinet (C.B-17 SCID mice) (12:12 h light–dark cycle, water and food ad libitum, 22–24 °C). All appropriate measures were taken to minimize pain or discomfort of animals. All the experimental procedures (induction of colitis, daily monitoring of the disease) and euthanasia by CO_2_ inhalation were performed between 9 a.m. and 2 p.m. All animal experiments were carried out according to the guidelines for the use and care of laboratory animals and they were authorized by the local Animal Care Committee “Organismo Preposto al Benessere degli Animali” and the Italian Ministry of Health “Ministero della Salute” (Authorization n.853/2021).

### 4.2. Development of Colitis

To induce colitis, pure naïve CD4^+^CD25^−^ T cells were adoptively transferred from healthy syngeneic Balb/c mice to C.B.-17 SCID recipients. CD4^+^CD25^−^ T cells were negatively selected from splenocytes and inguinal lymph nodes of Balb/c mice through CD4^+^CD25^+^ regulatory T cell isolation kit (MACS Milteny Biotec, Bergisch Gladbach, Germany). The purity of the cells was checked by flow cytometry before reconstitution (>98% of the CD4^+^ cells were CD25^−^). The recipients were reconstituted by intraperitoneal (i.p.) infusion of 10^6^ cells in 300 μL sterile PBS 100 mM and immediately moved from the SPF ventilated cabinet to a normal housing environment and non-autoclaved food. C.B.-17 SCID mice i.p. infused with 300 μL sterile PBS, containing 10^6^ CD4^+^ T cells, represented sham mice.

### 4.3. Experimental Design

As showed by Figure 8, pharmacological treatments started 7 days after i.p. infusion of CD4^+^ T cells (week 0) and were applied twice daily, 8 h apart, by subcutaneous (s.c.) administration of the vehicle or the drugs for the following 8 weeks until sacrifice by means of CO_2_ inhalation.

Mice were assigned through block randomization to the sham group (S; *n* = 6), s.c. treated with 10 mL/kg vehicle (DMSO 1% in saline solution), or to the following experimental groups of colitic mice: control (C) (vehicle, 10 mL/kg; *n* = 8), **MR120** 1 mg/kg (*n* = 8), or **MR452** 1 mg/kg (*n* = 8).

The dosage of **MR120** was chosen on the basis of Allodi et al., 2023 [15]; a similar dosage was applied for its derivative **MR452**. The study was performed using experimental blocks composed of 8 mice that were randomly assigned to S, C, **MR120** and **MR452** groups of treatments, each one encompassing 2 animals.

### 4.4. Evaluation of Inflammatory Responses

Body weight, stools consistency, and rectal bleeding were examined and registered weekly throughout the experimentation by unaware observers, in order to assess the Disease Activity Index (DAI). Immediately after sacrifice, the macroscopic damage of colonic mucosa was assessed as macroscopic score after visual inspection. Colon, lungs, spleen and mesenteric lymph nodes were collected for subsequent microscopic, biochemical, or flow cytometry analyses.

#### 4.4.1. Disease Activity Index (DAI)

The DAI, expressing the severity of the disease, is calculated as the total score resulting from body weight loss, rectal bleeding, and stool consistency according to Cooper’s modified method [33] (maximal value = 9).

The scores were quantified as follows:○Stool consistency: 0 (normal), 1 (soft), and 2 (liquid);○Rectal bleeding: 0 (no bleeding), 1 (light bleeding), and 2 (heavy bleeding);○Body weight loss: 0 (<5%), 1 (5–10%), 2 (10–15%), 3 (15–20%), 4 (20–25%), and 5 (>25%).

#### 4.4.2. Colon Length and Thickness

To evaluate the deposition of fibrotic material induced by the inflammatory state, colon length and weight were measured, and the thickness was estimated by calculating the ratio weight (mg)/length (cm) [34].

#### 4.4.3. Macroscopic Score

The colon was explanted, opened longitudinally, flushed with saline solution and macroscopic score was immediately evaluated through inspection of the mucosa, performed by two investigators unaware of the treatments applied. Macroscopic score was determined according to previously published criteria [35], as the sum of scores (max = 14) attributed as follows: presence of strictures and hypertrophic zones (0, absent; 1, 1 stricture; 2, 2 strictures; 3, more than 2 strictures); mucus (0, absent; 1, present); adhesion areas between the colon and other intra-abdominal organs (0, absent; 1, 1 adhesion area; 2, 2 adhesion areas; 3, more than 2 adhesion areas); intraluminal hemorrhage (0, absent; 1, present); erythema (0, absent; 1, presence of a crimsoned area < 1 cm^2^; 2, presence of a crimsoned area > 1 cm^2^); and ulcerations and necrotic areas (0, absent; 1, presence of a necrotic area < 0.5 cm^2^; 2, presence of a necrotic area ≥ 0.5 cm^2^ and <1 cm^2^; 3, presence of a necrotic area ≥ 1 cm^2^ and <1.5 cm^2^; 4, presence of a necrotic area ≥ 1.5 cm^2^).

#### 4.4.4. Spleen/Body Weight Ratio

Spleen/body weight ratio was determined as a marker of systemic inflammation. Following sacrifice, the spleen was weighed and the weight was normalized with respect to animal body weight (BW): the ratio was expressed as spleen (mg) × 1000/BW (g).

#### 4.4.5. Colon and Lung MPO Activity

MPO activity, marker of tissue granulocytic infiltration, was determined according to Ivey’s modified method in colon and lungs [36]. After being weighed, each colonic or lung sample was homogenized in ice-cold 0.02 M sodium phosphate buffer (pH 4.7), containing 0.015 M Na_2_EDTA and 1% Halt Protease Inhibitor Cocktail (ThermoFisher Scientific, Waltham, MA, USA), and centrifuged for 20 min at 12,500 RCF at 4 °C. Pellets were re-homogenized in 4 volumes of ice-cold 0.2 M sodium phosphate buffer (pH 5.4) containing 0.5% hexadecylthrimethyl-ammonium bromide (HTAB) (Sigma-Aldrich™, St. Louis, MO, USA) and 1% Halt Protease Inhibitor Cocktail (ThermoFisher Scientific, Waltham, MA, USA). Samples were then subjected to 3 cycles of freezing and thawing and centrifuged for 30 min at 15,500 RCF at 4 °C. An amount of 50 µL of the supernatant was then allowed to react with 950 µL of 0.2 M sodium phosphate buffer, containing 1.6 mM tetramethylbenzidine, 0.3 mM H_2_O_2_, 12% dimethyl formamide, and 40% Dulbecco’s phosphate-buffered saline (PBS). Each assay was performed in duplicate and the rate of change in absorbance was measured spectrophotometrically at 690 nm (TECAN Sunrise™ powered by Magellan™ Standard software data analysis, Mannedorf, Switzerland). An amount of 1 unit of MPO was defined as the quantity of enzyme degrading 1 μmol of peroxide per minute at 25 °C. Data were normalized with edema values and expressed as U/g of dry weight tissue.

#### 4.4.6. Flow Cytometry Assays

##### Isolation of Splenocytes

After sacrifice, the spleen was removed, mechanically dispersed through a 100 μm cell-strainer, and washed with PBS containing 0.6 mM EDTA (PBS-EDTA). The cellular suspension was then centrifuged at 1500 RCF for 10 min at 4 °C, the pellet re-suspended in PBS-EDTA, incubated with 2 mL of NH_4_Cl lysis buffer (0.15 M NH_4_Cl, 1 mM KHCO_3_, 0.1 mM EDTA in distilled water) for 5 min, in the dark, to provoke erythrocytes lysis and centrifuged at 1500 RCF for 10 min at 4 °C. Then, pellets were washed with PBS-EDTA and re-suspended in 5 mL cell staining buffer (PBS containing 0.5% fetal calf serum (FCS) and 0.1% sodium azide). Finally, the cellular suspension was stained with fluorescent antibodies [37].

##### Isolation of Mesenteric Lymph Nodes (MLNs)

After sacrifice, harvesting of the whole MLN chain located in the mesentery of proximal colon was performed. The explanted tissue was rinsed with PBS, vascular and adipose tissues were removed to isolate MLN, mechanically dispersed through a 100 μm cell-strainer and washed with Hank’s Balanced Salt Solution (HBSS) containing 5% FCS. The obtained suspension was centrifuged at 1500 RCF for 10 min at 4 °C, the pellet was washed with HBSS containing 5% FCS and re-suspended in 3 mL cell staining buffer. Finally, the cellular suspension was stained with fluorescent antibodies.

##### Immunofluorescent Staining

Prior to staining with antibodies, 200 µL of cellular suspension was incubated with IgG1-Fc (1 µg/10^6^ cells) for 10 min in the dark at 4 °C to block non-specific binding sites for antibodies. The following antibodies were used for fluorescent staining: phycoerythrin-cyanine 5 (PE-Cy5) conjugated anti-mouse CD3ε (0.25 µg/10^6^ cells, catalog number 15-0031, lot number B226301), fluorescein isothiocyanate (FITC) anti-mouse CD4 (0.25 µg/10^6^ cells, catalog number 100406, lot number B210488) (BioLegend™, San Diego, CA, USA). Cells were incubated with antibodies for 1 h in the dark at 4 °C, washed with PBS to remove excessive antibody and suspended in cell staining buffer to perform flow cytometry analysis. The viability of the cellular suspension was determined through propidium iodide (PI) (BioLegend™, San Diego, CA, USA) staining, a membrane impermeable fluorescent dye, excluded by viable cells, that emits red fluorescence by binding to DNA, thus resulting as a suitable marker for dead cells. Cells were incubated with PI 10 µg/mL for 1 min in the dark, at room temperature, and immediately subjected to flow cytometry analysis. After identification of singlets using Forward Scatter Area (FSC-A) vs. Forward Scatter Height (FSC-H), only PI^-ve^ cells were included in the analysis (Appendix A).

Samples were analyzed using InCyte™ software, version 2.6 (Merck Millipore, Darmstadt, Germany). Cell populations were defined as follows: lymphocytes gated in the FSC-SSC plot (FSC low/SSC low) (Appendix A); T lymphocytes (CD3^+^ lymphocytes); and CD4^+^ T lymphocytes (CD3^+^CD4^+^ lymphocytes). Percentages of CD4^+^ T lymphocytes with respect to CD3^+^ lymphocytes were calculated.

#### 4.4.7. Analysis of Inflammatory Cytokines

##### ELISA Assays

After homogenization in PBS containing a protease inhibitor cocktail of 10 μL/mL and EDTA 5 mM, colonic tissue was frozen–thawed for 10 min in liquid nitrogen and 15 min in a 37 °C water bath for 2 cycles, and the samples were then centrifuged at 5000 rpm for 5 min at 4 °C. The supernatant, containing the inflammatory cytokines CCL20, IL-6 and IL-1β, was assayed by means of available commercial ELISA kits [CCL20: Mouse MIP-3a ELISA Kit (Wuhan Fine Biotech Co., Ltd.; Wuhan, Cina); IL-6: Mouse IL-6 ELISA Kit (RayBiotech, Peachtree Corners, GA, USA); IL-1β: Mouse Interleukin 1β ELISA Kit (Cusabio, Houston, TX, USA)]. The absorbance was measured spectrophotometrically at 450 nm and expressed as ng/g of proteins present in each sample. The protein concentration of each sample was determined through the bicinchoninic acid (BCA) protein assay kit (ThermoFisher Scientific Inc., Waltham, MA, USA). Briefly, 15 μL of supernatant was made to react with 200 μL of BCA mix reagent and, after an incubation of 30 min, the absorbance was determined spectrophotometrically at 550 nm.

##### Reverse Transcription Polymerase Chain Reaction (RT-PCR)

Total RNA was extracted from murine colons using Qiagen RNeasy Protect Mini Kit (Qiagen, Hilden, Germany) and quantified using a NanoQuant PlateTM in an Infinite M Nano Machine (TECAN, Männedorf, Switzerland). An amount of 1 μg of RNA was reverse transcribed into complementary DNA (cDNA) and amplified using OneStep RT-PCR kit (Qiagen, Hilden, Germany), according to the manufacturer’s protocol. The following primers, purchased from Life Technologies (ThermoFisher Scientific Inc., Waltham, MA, USA), were used:○HPRT:

TCAGTCAACGGGGGACATAAA (sense)

GGGGCTGTACTGCTTAACCAG (antisense)

○CCL20:

TCTTGACTCTTAGGCTGAGG (sense)

CAGAAGCAGCAAGCAACTAC (antisense)

○IL-17:

CCCTGGACTCTCCACCGCAA (sense)

TCCCTCCGCATTGACACAGC (antisense)

○INF-γ:

TGAACGCTACACACTGCATCTTGG (sense)

CGACTCCTTTTCCGCTTCCTGAG (antisense)

All constructs were amplified applying the thermal cycler conditions suggested by QIAGEN OneStep RT-PCR Handbook (Qiagen, Hilden, Germany). PCR products were separated on 2% agarose gels and acquired with ChemiDoc Imaging System following RedSafe staining and analyzed by Image Lab software, version 5.0. Finally, mRNA levels were normalized with respect to the levels of hypoxanthine phosphoribosyl-transferase (*HPRT*) mRNA used as an internal control.

### 4.5. Statistics

All data were presented as mean ± SEM. Comparison among experimental groups was made using analysis of variance (one-way or two-way ANOVA), followed by Bonferroni’s post-test, a conservative post-test commonly applied to groups with equal size, when *p* < 0.05, chosen as the level of statistical significance, was achieved. Non-parametric Kruskal–Wallis analysis, followed by Dunn’s post-test, was applied for statistical comparison of macroscopic score. All analyses were performed using Prism 9 software (GraphPad Software Inc., San Diego, CA, USA).

### 4.6. Materials

Benzofuran-2-carboxamide derivatives MR120 and MR452 were synthesized and characterized by us as reported, respectively, in [14,17]. They were dissolved in saline solution containing DMSO 1% the day of the experiment. All reagents and kits are listed as follows:CD4^+^CD25^+^ regulatory T cell isolation kit-cod. 130-091-041 (MACS Milteny Biotec, Bergisch Gladbach, Germany).Halt Protease Inhibitor Cocktail 100x-cod. 78429 (ThermoFisher Scientific, Waltham, MA, USA).Hexadecylthrimethyl-ammonium bromide (HTAB) cod. H5882 (Sigma-Aldrich™, St. Louis, MO, USA).Phycoerythrin-cyanine 5 (PE-Cy5) conjugated anti-mouse CD3ε (catalog number 15-0031, lot number B226301); fluorescein isothiocyanate (FITC) anti-mouse CD4 (catalog number 100406, lot number B210488); propidium iodide were purchased from BioLegend™ (San Diego, CA, USA).IgG1-Fc was purchased from Millipore™ (Merck, Darmstadt, Germany).Hank’s Balanced Salt Solution (HBSS) and fetal calf serum (FCS) were purchased from Euroclone (Milano, Italy).Mouse MIP-3a ELISA Kit, cod. EM1210 (Wuhan Fine Biotech Co., Ltd.; Wuhan, Cina).Mouse IL-6 ELISA Kit, cod. 126ELM-IL6-CL-1 (RayBiotech, Peachtree Corners, GA, USA).Mouse Interleukin 1β ELISA Kit, cod. E08054 (Cusabio, Houston, TX, USA).Bicinchoninic acid (BCA) protein assay kit, cod. 23225 (ThermoFisher Scientific Inc., Waltham, MA, USA).Qiagen RNeasy Protect Mini Kit, cod. 74104 (Qiagen, Hilden, Germany).OneStep RT-PCR kit, cod. 210212 (Qiagen, Hilden, Germany).Primers were purchased from Life Technologies, ThermoFisher Scientific Inc., Waltham, MA, USA.Hydrogen peroxide, tetramethylbenzidine, dimethyl formamide, EDTA, and all the other reagents were purchased from Sigma-Aldrich™, St. Louis, MO, USA.

## Figures and Tables

**Figure 1 pharmaceuticals-18-01327-f001:**
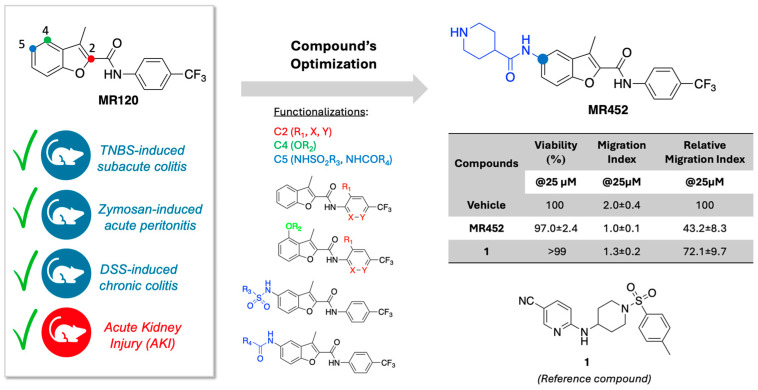
Compound **MR120** and its optimized derivative **MR452**. The data reported in the table refer to [18], where compound MR452 was reported as **25f**. The reference compound **1** was originally reported by Takeda Pharmaceutical [17].

**Figure 2 pharmaceuticals-18-01327-f002:**
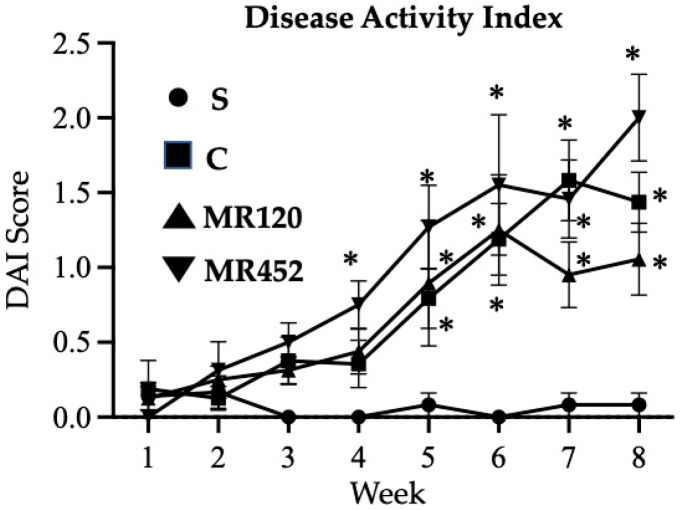
Curves representing the Disease Activity Index (DAI) score, assessed during adoptive transfer colitis in vehicle-treated sham mice (S •) and colitic mice administered with vehicle (C ■), **MR120** 1 mg/kg (▲) or **MR452** 1 mg/kg (▼) (*n* = 6–8 independent values per group). * *p* < 0.05 vs. S mice, two-way ANOVA followed by Tukey’s post-test.

**Figure 3 pharmaceuticals-18-01327-f003:**
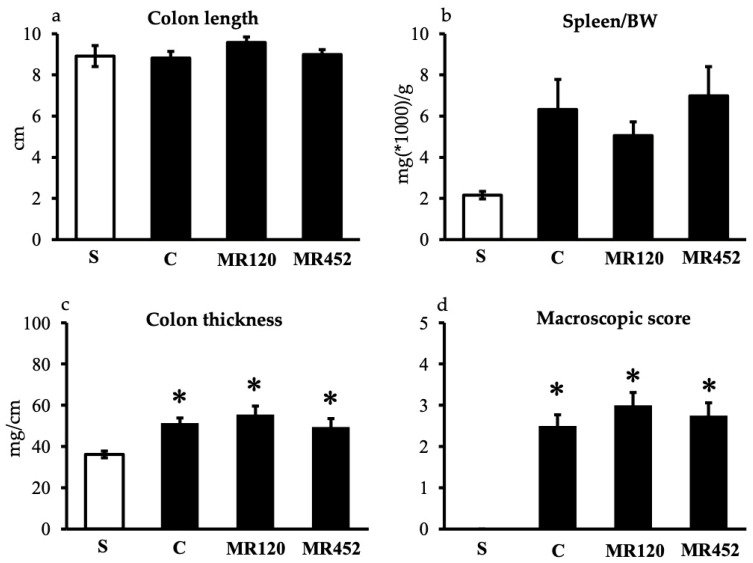
Effects of **MR120** and **MR452** on colon length (**a**), spleen/body weight (BW) ratio (**b**), colon thickness (**c**), and macroscopic score (**d**) assessed in vehicle-treated S mice and in CD4^+^CD25^−^ T cell transferred mice administered with vehicle (C), **MR120** 1 mg/kg or **MR452** 1 mg/kg (*n* = 6–8 independent values per group). * *p* < 0.05 vs. S mice, one-way ANOVA followed by Bonferroni’s post-test; Kruskal–Wallis followed by Dunn’s post-test (for macroscopic score).

**Figure 4 pharmaceuticals-18-01327-f004:**
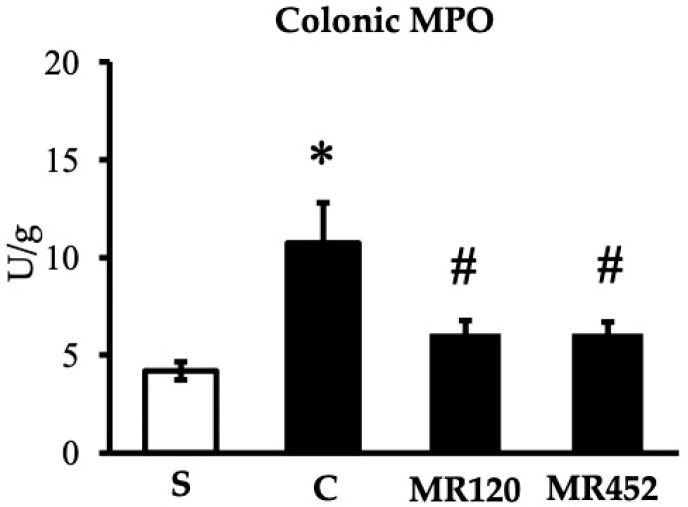
Effects of **MR120** and **MR452** on colonic MPO activity (U/g dry colon) assessed in vehicle-treated S mice and in CD4^+^CD25^−^ T cell transferred mice administered with vehicle (C), **MR120** 1 mg/kg or **MR452** 1 mg/kg (*n* = 6–8 independent values per group). * *p* < 0.05 vs. S mice, # *p* < 0.05 vs. C mice, one-way ANOVA followed by Bonferroni’s post-test.

**Figure 5 pharmaceuticals-18-01327-f005:**
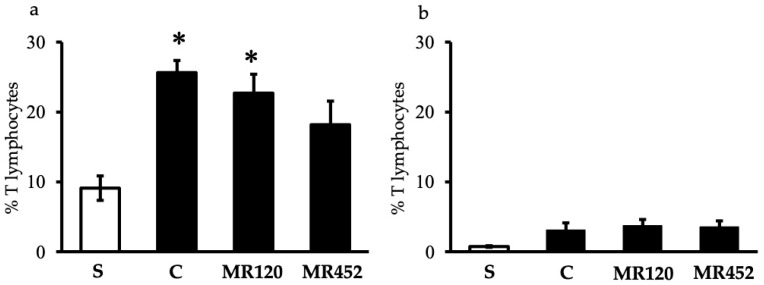
Effects of **MR120** and **MR452** on T lymphocytes of mesenteric lymph nodes (**a**) and spleen (**b**) in vehicle-treated normal mice (S) and colitic mice administered with vehicle (C), **MR120** 1 mg/Kg or **MR452** 1 mg/kg (*n* = 5–8 independent values per group). * *p* < 0.05 vs. S; one-way ANOVA + Bonferroni’s post-test.

**Figure 6 pharmaceuticals-18-01327-f006:**
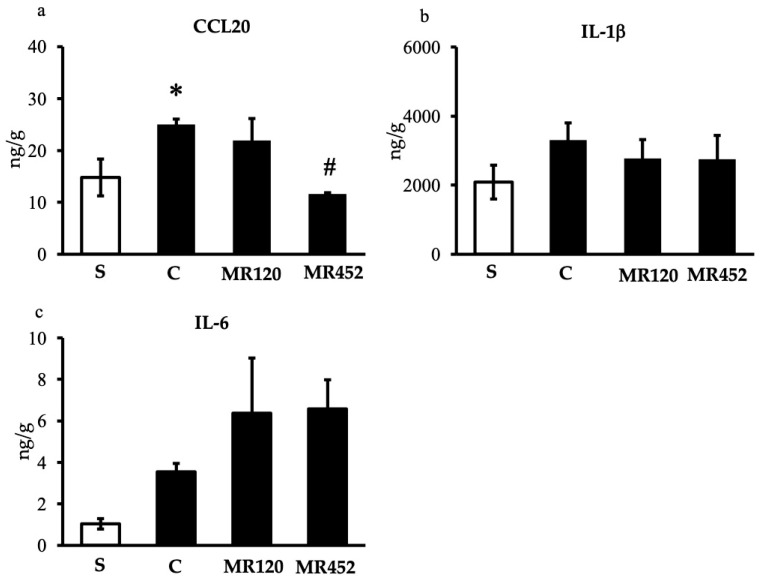
Histograms representing colonic CCL20 (**a**), IL-1β (**b**), and IL-6 (**c**) levels in vehicle-treated normal mice (S) and colitic mice administered with vehicle (C), **MR120** 1 mg/kg or **MR452** 1 mg/kg (*n* = 4–7 independent values per group). * *p* < 0.05 vs. S; # *p* < 0.05 vs. C, one-way ANOVA + Bonferroni’s post-test.

**Figure 7 pharmaceuticals-18-01327-f007:**
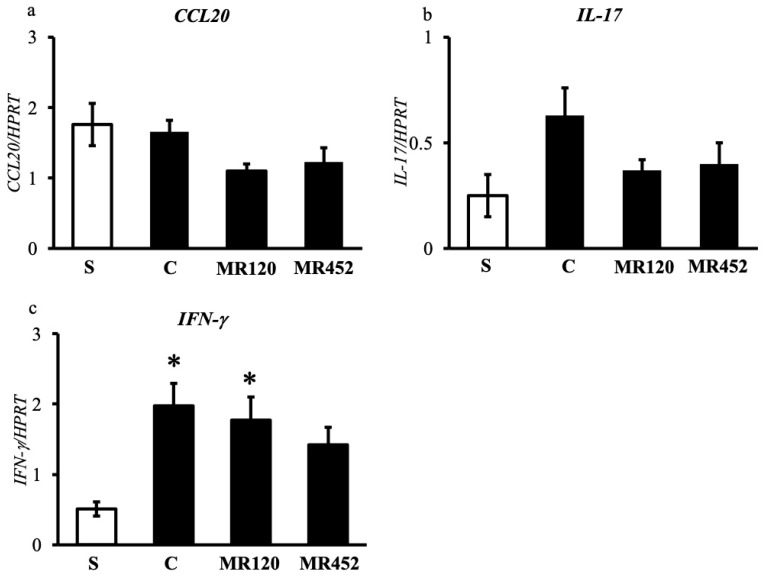
Histograms representing the quantification of colonic mRNA levels of *CCL20* (**a**), *IL-17* (**b**), and *IFN-γ* (**c**) in vehicle-treated normal mice (S) and colitic mice administered with vehicle (C), **MR120** 1 mg/kg or **MR452** 1 mg/kg. In each sample, mRNA levels were normalized with respect to the levels of hypoxanthine phosphoribosyl-transferase (*HPRT*) mRNA used as internal control (*n* = 4–6 independent values per group). * *p* < 0.05 vs. S, one-way ANOVA + Bonferroni’s post-test.

**Figure 8 pharmaceuticals-18-01327-f008:**
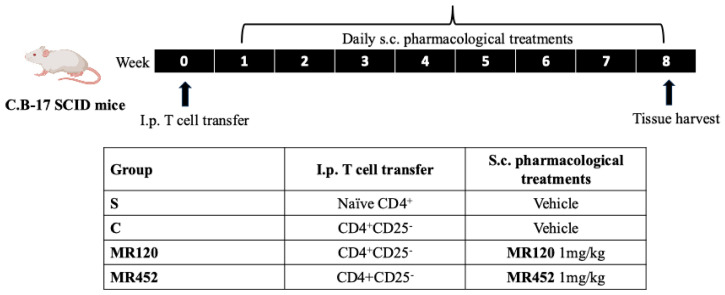
Experimental design for adoptive T cell transfer colitis and pharmacological treatments.

**Table 1 pharmaceuticals-18-01327-t001:** Percentage of CD4^+^ T lymphocytes in MLNs obtained from vehicle-treated normal mice (S) and colitic mice administered with vehicle (C), **MR120** 1 mg/kg or **MR452** 1 mg/kg (*n* = 5–8 independent values per group).

	S	C	MR120	MR452
%CD4^+^ T lymphocytes	80.5 ± 3.4	87.4 ± 2.1	87.5 ± 2.8	77.5 ± 3.4

## Data Availability

The data are available upon request.

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
