# Peer review of "Effects of CCL20/CCR6 Modulators in a T Cell Adoptive Transfer Model of Colitis"

_pharmaceuticals, 2025, doi:10.3390/ph18091327_

Round 1
Reviewer 1 Report
Comments and Suggestions for Authors
The manuscript titled "Effects of CCL20/CCR6 Modulators in a T Cell Adoptive Transfer Model of Colitis" evaluates the efficacy of two CCR6 antagonists (MR120 and MR452) in a T cell transfer-induced colitis model, extending previous positive findings from other IBD models (TNBS/DSS). While the study is methodologically robust and targets the relevant CCL20/CCR6 axis, the predominantly negative results raise critical concerns regarding the suitability of the model, the effectiveness of the compounds in this specific context, and the underlying mechanistic interpretation. Substantial revisions are necessary to address these issues prior to publication.
Major concerns:
- The T cell transfer model was selected due to its reliance on the Treg/Th17 imbalance (lines 190-200). However, existing literature consistently demonstrates that CCR6 deficiency (via gene knockout) in donor T cells aggravates colitis in this model (lines 224-228). This raises a critical contradiction: why anticipate pharmacological CCR6 antagonism to provide protection when genetic ablation intensifies disease severity? Moreover, while the authors briefly acknowledge this conflict (lines 228-231), they dismiss it without offering a robust mechanistic explanation or hypothesis-driven analysis. The rationale for employing this model to evaluate CCR6 antagonists is weakly substantiated and potentially flawed. Although the discussion (lines 233-249) speculates on factors such as timing and disease progression, it fails to resolve the underlying contradiction. The authors must thoroughly address this paradox.
- The data clearly demonstrate limited efficacy of MR120 and MR452 in this model. The only notable positive outcome was a reduction in MPO activity. Meanwhile, the Abstract, Introduction, and Discussion place strong emphasis on the therapeutic potential of CCR6 antagonism based on findings from previous models. While the current negative results in this specific model are underplayed. The authors should clearly state the limited efficacy identified in this study within both the Abstract and Discussion sections. Ensure conclusions remain strictly aligned with the presented data, avoiding any overstating.
- The core hypothesis centers on the Treg/Th17 imbalance; however, no data on Treg or Th17 cells in target tissues are provided. Flow cytometry analysis only includes total CD3+ and CD3+CD4+ cells (Fig. 5, Table 1). Measuring IL-17 protein/mRNA levels (Fig. 6b, 7b) is insufficient, as it fails to identify cellular sources or determine ratios. Please add data on FoxP3+ Tregs and RORγt+ (or IL-17+) Th17 cells in MLNs and/or the colon. Additionally, evaluate the Treg/Th17 ratio. This is essential for validating the central hypothesis of the study and the appropriateness of the chosen model.
- The group sizes (n=5-8) are relatively small for a model with variable disease progression. Negative findings, such as the lack of DAI improvement, may stem from insufficient statistical power rather than genuine absence of effect. Clarify whether power calculations were conducted to assess adequacy of sample size.
- Section 4.4.6 requires additional clarification and detail:
- Specify the method used for lymphocyte gating, including an example of the FSC/SSC plot.
- Indicate the viability threshold applied. For instance, PI exclusion criteria.
- Include representative gating plots in the supplementary data for better transparency.
- Explain the omission of spleen T cell data (line 131: "data not shown").
Minor concerns:
- Figure 2: Label the Y-axis as "DAI Score".
2.Figure 3: Ensure statistical annotations on the graphs are clear and easily interpretable.
- Figure 7: Update the Y-axis label "Fold Change" to include a reference, such as "Fold Change vs. Sham" or specify the control group for clarity.
- Figure 8: Replace the term "Suppression" terms with "Sacrifice" or "Euthanasia". Additionally, clarify the timeline. For xampl: confirm if Day 0 refers to cell transfer).
Writing and Clarity:
- Line 81/82: The definition of "Relative migration index" appears misplaced; it should be referenced in the previous study ([17]).
Line 174: Revise "The current work aims at applying..." to "This study aimed to apply..." for proper past tense usage.
Line 231: "A modest anti-inflammatory effect was observed" is overstated; clarify that only MPO reduction was noted.
Line 255: Correct "hypotesize" to "hypothesize."
Line 411: Adjust "Reverse Transcription Polymerase Chain Reaction (RT-PCR)" to "Quantitative RT-PCR (qRT-PCR)".
Author Response
Response to reviewer 1
We would like to thank Reviewer 1 for the helpful and very thoughtful comments on the manuscript.
We did the appropriate changes, highlighted in yellow, and a point-by-point response can be found below.
Major concerns
Q1: The T cell transfer model was selected due to its reliance on the Treg/Th17 imbalance (lines 190-200). However, existing literature consistently demonstrates that CCR6 deficiency (via gene knockout) in donor T cells aggravates colitis in this model (lines 224-228). This raises a critical contradiction: why anticipate pharmacological CCR6 antagonism to provide protection when genetic ablation intensifies disease severity? Moreover, while the authors briefly acknowledge this conflict (lines 228-231), they dismiss it without offering a robust mechanistic explanation or hypothesis-driven analysis. The rationale for employing this model to evaluate CCR6 antagonists is weakly substantiated and potentially flawed. Although the discussion (lines 233-249) speculates on factors such as timing and disease progression, it fails to resolve the underlying contradiction. The authors must thoroughly address this paradox.
R1: We thank the reviewer for the observations. It is possible that different phenotypes may ensue from targeting a protein function either via small molecule inhibitors or gene knock out (Knight and Shokat, 2007). In our case, the fact that CCR6 deficiency via gene knockout in donor T cells aggravates colitis in the T cell adoptive transfer model does not automatically exclude the potential advantages of CCR6 antagonists in colitis. It is possible that both genetic deletion and pharmacological treatment with MR120 and MR452 might exert different off-target effects, or that genetic deletion might lead to compensatory changes to other genes during mice development that could affect the role of CCR6 receptor in intestinal inflammation in adult mice and do not correspond to the in vivo effects of CCR6 inhibitors. These considerations have been introduced in the Discussion, p. 9, lines 261-274.
Q2: The data clearly demonstrate limited efficacy of MR120 and MR452 in this model. The only notable positive outcome was a reduction in MPO activity. Meanwhile, the Abstract, Introduction, and Discussion place strong emphasis on the therapeutic potential of CCR6 antagonism based on findings from previous models. While the current negative results in this specific model are underplayed. The authors should clearly state the limited efficacy identified in this study within both the Abstract and Discussion sections. Ensure conclusions remain strictly aligned with the presented data, avoiding any overstating.
R2: As suggested by the reviewer, Abstract and Discussion have been re-phrased to tone down the emphasis, remaining aligned to the obtained results.
Q3: The core hypothesis centers on the Treg/Th17 imbalance; however, no data on Treg or Th17 cells in target tissues are provided. Flow cytometry analysis only includes total CD3+ and CD3+CD4+ cells (Fig. 5, Table 1). Measuring IL-17 protein/mRNA levels (Fig. 6b, 7b) is insufficient, as it fails to identify cellular sources or determine ratios. Please add data on FoxP3+ Tregs and RORγt+ (or IL-17+) Th17 cells in MLNs and/or the colon. Additionally, evaluate the Treg/Th17 ratio. This is essential for validating the central hypothesis of the study and the appropriateness of the chosen model.
R3: We agree with the reviewer’s criticism. We did not have the possibility to perform the measurements required for the quantification of Foxp3+ and RORgt+ cells in murine colon and MLNs. Since we can only presume that Treg/Th17 imbalance occurs in the experimental model we applied, based on the infusion of CD4+CD25- cells to congenic SCID mice, the weight given to these two CD4+ subtypes has been attenuated, and the aim and discussion have been re-phrased to better comply with the rational of the study and the reviewer’s comments.
Q4: The group sizes (n=5-8) are relatively small for a model with variable disease progression. Negative findings, such as the lack of DAI improvement, may stem from insufficient statistical power rather than genuine absence of effect. Clarify whether power calculations were conducted to assess adequacy of sample size.
R4: The sample size was assessed via G-Power analysis, considering an a error of 0.05, (1-b) error of 0.8, 4 experimental groups and the effect size for MPO value of 0.5. The calculated number of mice composing each experimental group, according to the protocol approved by the local ethical committee, was 10. Actually, we reduced to 6 units the sham group, given its lower variability, and to 8 the number of experimental units in the colitis groups. Since we aimed to determine from each mouse not only MPO activity, but also cytokine levels and RT-PCR data, and since the colon tissue obtained from each animal was not sufficient to determine concurrently all the parameters, we had to reduce the group sizes to obtain data for all parameters, giving in any case precedence to MPO activity assessment.
Q5: Section 4.4.6 requires additional clarification and detail:
- Specify the method used for lymphocyte gating, including an example of the FSC/SSC plot.
R.: As reported in the manuscript (Section 4.4.6.3, p. 13, lines 446-50), “Samples were analysed using InCyteÔ software (Merck Millipore, Darmstadt, Germany). Cell populations were defined as follows: lymphocytes gated in the Forward Scatter (FSC)-Side Scatter (SSC) plot (FSC low: SSC low) (Figure S2); T lymphocytes (CD3+ lymphocytes); CD4+ T lymphocytes (CD3+CD4+ lymphocytes). Percentages of CD4+ T lymphocytes with respect to CD3+ lymphocytes were calculated”.
- Indicate the viability threshold applied. For instance, PI exclusion criteria.
R.: As indicated in the manuscript (Section 4.4.6.3, p. 13, lines 438-45), “The viability of the cellular suspension was determined through propidium iodide (PI) staining, a membrane impermeable fluorescent dye, excluded by viable cells, that emits red fluorescence by binding to DNA, thus resulting as a suitable marker for dead cells. Cells were incubated with PI 10 µg/mL for 1 minute in the dark, at room temperature, and immediately subjected to flow cytometry analysis. Only PI-ve cells were included in the analysis (Figure S1).”
- Include representative gating plots in the supplementary data for better transparency.
R.: We apologize for the inconvenience but, at the moment and for the upcoming months, our labs are condemned, because of seismic retrofitting and improvement works deliberated by the governing body of our University. Unfortunately, we do not have the possibility to retrieve representative gating plots since flow cytometry apparatus is currently not available.
- Explain the omission of spleen T cell data (line 131: "data not shown").
R.: As required by the reviewer, spleen T cell data have been introduced in section 2.2 and a novel figure has been added (Figure 5b).
Minor concerns
Q1: Figure 2: Label the Y-axis as "DAI Score".
R1: The Y-axis of Figure 2 has been properly revised.
Q2: Figure 3: Ensure statistical annotations on the graphs are clear and easily interpretable.
R2: We have increased statistical annotations on graphs to make them clearer and more easily understandable.
Q3: Figure 7: Update the Y-axis label "Fold Change" to include a reference, such as "Fold Change vs. Sham" or specify the control group for clarity.
R3: As indicated in the manuscript (p. 14; lines 491-3), the reference is represented by mRNA levels of hypoxanthine phosphoribosyl-transferase (HPRT) mRNA, used as internal control for each each sample.
Q4: Figure 8: Replace the term "Suppression" terms with "Sacrifice" or "Euthanasia". Additionally, clarify the timeline. For xampl: confirm if Day 0 refers to cell transfer).
R3: The term “suppression” has been replaced by “sacrifice”. As added in the manuscript, week 0 corresponds to the week of i.p. CD4+ transfer.
Writing and clarity
Q5: Line 81/82: The definition of "Relative migration index" appears misplaced; it should be referenced in the previous study ([17]).
R5: As suggested by the reviewer, the definition has been removed and the reference has been introduced in the caption
Line 174: Revise "The current work aims at applying..." to "This study aimed to apply..." for proper past tense usage.
R.: The sentence has been corrected.
Line 231: "A modest anti-inflammatory effect was observed" is overstated; clarify that only MPO reduction was noted.
Line 255: Correct "hypotesize" to "hypothesize."
R.: The suggested corrections were introduced
Line 411: Adjust "Reverse Transcription Polymerase Chain Reaction (RT-PCR)" to "Quantitative RT-PCR (qRT-PCR)".
R.: In this investigation we did not apply “Quantitative RT-PCR (qRT-PCR)” but “Reverse Transcription Polymerase Chain Reaction (RT-PCR)"

Reviewer 2 Report
Comments and Suggestions for Authors
The manuscript entitled “Effects of CCL20/CCR6 Modulators in a T Cell Adoptive Transfer Model of Colitis” is well written and presents important findings that are relevant to the field of immunology and inflammatory bowel disease. However, a few clarifications and additions are required to enhance the scientific rigor and reproducibility of the study. Please consider the following points:
-
Figure 1: Please use the original name of the reference compound (Cpd1) throughout the figure and legend. The current naming adds unnecessary confusion.
-
Figure 2: Clarify what is meant by "S mice" in the figure and legend. If it refers to saline-treated or control mice, please state this explicitly.
-
Figure 3: It is unclear whether there is a statistically significant difference in the spleen-to-body weight (S/BW) ratio between groups. Please include statistical comparisons in the figure and legend.
-
Experimental Design:
-
Please explain the rationale for using 1% DMSO in saline as the vehicle.
-
Clarify what is meant by "10 mg/kg vehicle." If this refers to the dose of the compound administered, please rephrase to avoid ambiguity.
-
-
Materials and Methods: All reagents and kits used should be listed along with their sources (company names and catalog numbers, where available). This includes cytokine kits, antibodies, and any proprietary formulations.
-
Statistical Analysis: Please justify the use of Bonferroni’s post-test in the context of your dataset. A brief explanation of its relevance for multiple comparisons would improve transparency.
With these additions and clarifications, the manuscript will be significantly strengthened and more suitable for publication.
Author Response
We would like to thank Reviewer 2 for the helpful and very thoughtful comments on the manuscript.
We did the appropriate changes, highlighted in yellow, and a point-by-point response can be found below.
Q1: Figure 1: Please use the original name of the reference compound (Cpd1) throughout the figure and legend. The current naming adds unnecessary confusion.
R1: The original name of the reference compound reported in the cited publication [17] is compound 1. Figure and legend have been corrected accordingly.
Q2: Figure 2: Clarify what is meant by "S mice" in the figure and legend. If it refers to saline-treated or control mice, please state this explicitly.
R2: In Figure 2, S mice correspond to sham mice, represented by C.B.-17 SCID mice i.p. infused with 300 ml sterile PBS, containing 106 CD4+ T cells, as indicated in section 4.2 of Materials and Methods.
Q3: Figure 3: It is unclear whether there is a statistically significant difference in the spleen-to-body weight (S/BW) ratio between groups. Please include statistical comparisons in the figure and legend.
R3: The difference among the spleen-to-body weight ratio of the various experimental groups is not significant and, therefore, statistical marks have not been provided.
Q4: Experimental Design:
Q.: Please explain the rationale for using 1% DMSO in saline as the vehicle
R.: 1% DMSO is considered a well-tolerated vehicle for in vivo administration to mice. According to Bartsch et al., 1976 (PMID 1036956), its LD50 is 6.2 ml/kg, corresponding to 0.124 ml for a mouse of 20g, a 60-times higher volume than that employed in our investigation as vehicle, represented by 1% DMSO 10ml/kg (i.e. 0.2ml/mouse).
Q.: Clarify what is meant by "10 mg/kg vehicle." If this refers to the dose of the compound administered, please rephrase to avoid ambiguity.
R.: We thank the reviewer for the request of clarification, but the vehicle was administered as a volume of 10ml/kg and not 10mg/kg, as reported in the manuscript, section 4.3, p.11, lines 338-41.
Q5: Materials and Methods: All reagents and kits used should be listed along with their sources (company names and catalog numbers, where available). This includes cytokine kits, antibodies, and any proprietary formulations.
R.: A list of the reagents and kits used has been introduced as section 4.6 in the manuscript.
Q6: Statistical Analysis: Please justify the use of Bonferroni’s post-test in the context of your dataset. A brief explanation of its relevance for multiple comparisons would improve transparency.
R.: As indicated in the manuscript (section 4.5), we commonly apply Bonferroni’s post-hoc test following parametric ANOVA test, for comparison of three or more groups with equal group size, as in the case of our dataset.

Reviewer 3 Report
Comments and Suggestions for Authors
The manuscript by Allodi et al., investigates the efficacy of two CCL20/CCR6 axis modulators, MR120 and its derivative MR452, in a T cell adoptive transfer model of colitis. The study aims to determine if these compounds, previously shown to be effective in other inflammatory bowel disease models, can ameliorate colitis. While the study provides valuable insights, its findings are largely modest, leading to important discussions about model suitability and mechanistic specificity.
Nevertheless, the manuscript represents an experimental structured investigation of the potential of MR120 and MR452 in the adoptive transfer colitis model. The authors are transparent about the limited efficacy observed in this specific model, which does not contribute much to the scientific understanding of these compounds and the complexity of IBD pathogenesis. Despite this, there are some issues:
Major:
-There is a significant weakness is the lack of a substantial positive effect on several primary indicators of colitis severity, particularly the DAI, colon thickness, and macroscopic inflammation score. While the reduction in MPO activity and CCL20 protein by MR452 are positive, they do not translate into overall clinical or macroscopic improvement in this model, raising questions about the compounds' broad utility in T-cell driven colitis.
-Why did the authors not examine a group with both modulators in the colitis model in question?
-How was the concentration of the modulators determined? Could it be that the concentration used is not effective in alleviating inflammation?
-The authors claim a TH1/Th2 shift. The abstract and discussion suggest that MR120 and MR452 "appear more clearly involved in the control of Th1 and Th2 adaptive immune responses" based on previous studies. However, the data presented in this manuscript for IFN-γ (a Th1 cytokine) mRNA levels show only a moderate increase in colitic mice and no "marked decrease" with treatment. To definitively support this conclusion within the context of this study, more direct evidence of Th1/Th2 modulation (e.g., flow cytometry of Th1/Th2 populations, broader cytokine profiles) in the adoptive transfer model would be beneficial.
-There ist additionally a lack of direct Treg/Th17 population data. Given that the adoptive transfer model is explicitly highlighted as being driven by "altered balance between Treg and Th17," more direct quantification of these specific T cell subsets in the colon and mesenteric lymph nodes (beyond just total T lymphocytes or CD4+ T lymphocytes) would have significantly strengthened the discussion around the compounds' impact on this crucial balance. While IL-17 mRNA was assessed, directly measuring Treg (Foxp3+) and Th17 (RORγt+) populations would be more informative.
-The manuscript correctly points out the different results of this model and earlier TNBS/DSS models for MR120. However, a more detailed investigation of why MPO reduction and CCL20 modulation by MR452 do not lead to better clinical benefits in this adoptive transfer model could be valuable. For example, is T cell-mediated inflammation so dominant that it masks the effects of reduced neutrophil infiltration or CCL20 levels?
Minor:
-Some figures, particularly those displaying molecular data (e.g., Figure 6b,c and Figure 7), have small text and somewhat crowded legends, which could be improved for better readability.
Despite the “modest” efficacy demonstrated in the adoptive transfer colitis model, this work makes a valuable contribution to the field. It serves as a reminder that the choice of preclinical model has a significant impact on the therapeutic effects observed and that agents that are effective in one inflammatory context are not necessarily directly transferable to others. The results suggest that MR120 and MR452 are more suitable for IBD manifestations.
In view of all this, the manuscript should be greatly improved and the statements and interpretations should be formulated more cautiously or supported by scientific data, which is why I suggest a major revision.
Author Response
We would like to thank Reviewer 3 for the helpful and very thoughtful comments on the manuscript.
We did the appropriate changes, highlighted in yellow, and a point-by-point response can be found below.
Major concerns
Q1: There is a significant weakness is the lack of a substantial positive effect on several primary indicators of colitis severity, particularly the DAI, colon thickness, and macroscopic inflammation score. While the reduction in MPO activity and CCL20 protein by MR452 are positive, they do not translate into overall clinical or macroscopic improvement in this model, raising questions about the compounds' broad utility in T-cell driven colitis.
R1: As correctly pointed out by the reviewer, the limited efficacy shown by CCR6 antagonists in the adoptive transfer model raises consistent doubts about their potential beneficial action (see p.9, lines 257-260; p. 10 p.277-9; p.10 lines 297-9) and also about the suitability of this experimental model to investigate the potential anti-inflammatory actions of CCR6 antagonists (see p. 10, lines 290-4; p. 10 lines 297-305).
Q2: Why did the authors not examine a group with both modulators in the colitis model in question?
R2: Given the modest in vivo characterization of MR120 and MR452 (the complete pharmacodynamic and pharmacokinetic properties of the molecules have not been defined) and the complex experimental model applied, we preferred to test them separately.
Q3: How was the concentration of the modulators determined? Could it be that the concentration used is not effective in alleviating inflammation?
R3: As reported in the manuscript, “The dosage of MR120 was chosen on the basis of Allodi et al. 2023 [15]; a similar dosage was applied for its derivative MR452.” As correctly suggested by the reviewer, it is definitely possible that the adopted dosage is lower than that required to obtain an effective anti-inflammatory action in the adoptive transfer model, as indicated in the manuscript at lines 280-1, “…a single dose was tested, while higher doses may produce stronger, either positive or negative, effects…”.
Q4: The authors claim a TH1/Th2 shift. The abstract and discussion suggest that MR120 and MR452 "appear more clearly involved in the control of Th1 and Th2 adaptive immune responses" based on previous studies. However, the data presented in this manuscript for IFN-γ (a Th1 cytokine) mRNA levels show only a moderate increase in colitic mice and no "marked decrease" with treatment. To definitively support this conclusion within the context of this study, more direct evidence of Th1/Th2 modulation (e.g., flow cytometry of Th1/Th2 populations, broader cytokine profiles) in the adoptive transfer model would be beneficial.
R4: We agree with the reviewer’s criticism and this conclusion has been removed from the abstract and the discussion.
Q5: There ist additionally a lack of direct Treg/Th17 population data. Given that the adoptive transfer model is explicitly highlighted as being driven by "altered balance between Treg and Th17," more direct quantification of these specific T cell subsets in the colon and mesenteric lymph nodes (beyond just total T lymphocytes or CD4+ T lymphocytes) would have significantly strengthened the discussion around the compounds' impact on this crucial balance. While IL-17 mRNA was assessed, directly measuring Treg (Foxp3+) and Th17 (RORγt+) populations would be more informative.
R4: We agree with the reviewer’s criticism. We did not have the possibility to perform the measurements required for the quantification of Foxp3+ and RORgt+ cells in murine colon and MLNs. Since we can only presume that Treg/Th17 imbalance occurs in the experimental model we applied, based on the infusion of CD4+CD25- cells to congenic SCID mice, the weight given to these two CD4+ subtypes has been attenuated, and the aim and discussion have been re-phrased to better comply with the rational of the study and the reviewer’s comments.
Q6: The manuscript correctly points out the different results of this model and earlier TNBS/DSS models for MR120. However, a more detailed investigation of why MPO reduction and CCL20 modulation by MR452 do not lead to better clinical benefits in this adoptive transfer model could be valuable. For example, is T cell-mediated inflammation so dominant that it masks the effects of reduced neutrophil infiltration or CCL20 levels?
R6: We thank the reviewer for the thoughtful observation and a new comment has been introduced in the Discussion section (p. 10, lines 299-305).
Minor concerns
Q.: Some figures, particularly those displaying molecular data (e.g., Figure 6b,c and Figure 7), have small text and somewhat crowded legends, which could be improved for better readability.
R.: As suggested by the reviewer, the size of Figures 6 and 7 has been increased.

Round 2
Reviewer 1 Report
Comments and Suggestions for Authors
The authors attempted to answer the queries appropriately.